

# TaGra: an open Python package for easily generating graphs from data tables through manifold learning

Davide Torre[1,2] and Davide Chicco[3,4]

[1] Luiss Guido Carli, Rome, Italy
[2] Istituto per le Applicazioni del Calcolo "Mauro Picone" (IAC), CNR (Consiglio Nazionale delle Ricerche), Rome, Lazio, Italy
[3] Dipartimento di Informatica Sistemistica e Comunicazione, University of Milan-Bicocca, Milan, Italy
[4] Institute of Health Policy Management and Evaluation, University of Toronto, Toronto, Ontario, Canada

## ABSTRACT

The challenge of analyzing high-dimensional data affects many scientific disciplines, from pharmacology to chemistry and biology. Traditional dimensionality reduction methods often oversimplify data, making it difficult to interpret individual points. This distortion can complicate the visualization of mutual distances between data points in the reduced space. Graphs provide an effective framework for representing objects and their relationships. One of their possible use is visualizing similarity patterns in tabular datasets. Here we introduce TaGra, an off-the-shelf package designed to generate a graph of similarity relations from tabular data. TaGra enables the visualization of datasets in 2D space, identification of typical data points and outliers, and assessment of the separation between items with different target variables. We describe TaGra's functionality, options and setup. The software including examples, instructions and a guide, is openly available on PyPI at https://pypi.org/project/TaGra/ and on GitHub at https://github.com/davidetorre92/TaGra.

# INTRODUCTION

High-dimensional data is pervasive across scientific disciplines, particularly in pattern recognition and machine learning (for example, *Giraud, 2021*, section 1.1). Such high-dimensional spaces often present a dual challenge: they contain substantial noise and redundant information while making it difficult to meaningfully visualize complex relationships between data points (*Altman & Krzywinski, 2018*).

Graph-based methods have gained significant traction in recent years due to their inherent ability to effectively capture and represent complex properties of networked data (*Carneiro & Zhao, 2018*). In analyzing high-dimensional datasets, the standard methodological approach typically follows a two-step process: (i) transforming feature vector data into a graph representation that preserves essential relationships, and

Corresponding author
Davide Torre, d.torre@iac.cnr.it

(ii) leveraging the resulting network structure to uncover and analyze underlying patterns, communities, and relationships (*Silva & Zhao, 2012*). This approach has proven particularly effective in maintaining both local and global data properties while providing intuitive visualizations of complex data structures. However, despite the proliferation of network analysis libraries, there remains a gap between general-purpose graph tools and specialized visualization solutions that integrate preprocessing, graph generation, and analysis in a cohesive workflow.

Multiple dimensionality reduction methodologies have been developed to address visualization challenges in high-dimensional spaces (*Nguyen & Holmes, 2019*). While methods like t-distributed Stochastic Neighbor Embedding (t-SNE) and Uniform Manifold Approximation and Projection (UMAP) have gained popularity for their ability to preserve local structures, they often sacrifice global relationships. Recent frameworks such as that proposed by *Mu, Goulermas & Ananiadou (2018)* incorporate both local and global structural constraints, but their specialized nature can limit general applicability across diverse datasets and analytical needs.

*Probst & Reymond (2019)* addressed scalability concerns in visualizing large high-dimensional datasets by developing TMAP (Tree MAP), a tree-based method capable of representing millions of data points while preserving global and local neighborhood structures. However, TMAP emphasizes scalability at the expense of comprehensive data preprocessing and analysis capabilities and graph generation versatility.

Here we present TaGra, an integrated open-source package that bridges the gap between dimensionality reduction techniques and network analysis. TaGra creates a graph of similarity relations from tabular data while providing tools to visualize data points in 2D space and quantify the separation between instances with different target variables.

A graph $G$ consists of the pair $(V, E)$, where $V$ are the vertices and $E$ is the set of relations connecting these vertices (*Newman, 2018*). The graph representation of a *tabular* dataset is effective for showing similarity relationships between data points. In particular, it can be used to reduce the size of a dataset while preserving the relationships between first neighbors (*Jia et al., 2022*).

TaGra provides a workflow where users can specify datasets and target variables through a minimal bash command, with additional customization available *via* JavaScript Object Notation (JSON) configuration. Our Python library integrates dimensionality reduction with comprehensive preprocessing capabilities—handling categorical variable encoding, missing data, and standardization automatically. Unlike existing approaches such as TMAP, TaGra emphasizes both visualization and analytical depth, incorporating detailed graph analyses including community detection and neighbours statistics.

The article is organized as follows: the Methods section describes TaGra's architecture and algorithms, including data preprocessing, graph creation, and analysis modules. The Results section demonstrates the application of TaGra to two medical informatics datasets, comparing different graph construction approaches and their performance in visualizing data relationships. The Discussion and Conclusions examines

limitations and potential applications of the package, while also comparing it with existing tools.

## METHODS

In this section, we will describe the architecture of the software, the parts of which it is composed and their dependencies, and go into detail on the methods by which we can obtain a graph from a dataframe and its visualisation.

With a minimal set of options, the software automates data preprocessing, including handling missing values and encoding categorical variables, and transforms tabular data into graphs. Its goal is to integrate and automate dataset preprocessing with graph creation and analysis, thus providing insights through community detection and degree distribution. Additionally, it offers detailed visualisations to enhance data interpretation and presentation.

### Software architecture

TaGra is designed with a modular architecture to streamline data preprocessing, graph creation, and graph analysis. The software consists of three main components: data preprocessing module, graph creation module and graph analysis module which are fully settable with a ready-to-use configuration file in JSON format. In Fig. 1 we show a pictorial representation of the entire package, complete with the main submodules.

We have used the following Python libraries:

(1) Preprocessing: Pandas (*Pandas Development Team, 2020*; *McKinney, 2010*) for the dataset handling, NumPy (*Harris et al., 2020*) for scaling the data and scikit-learn for the encoding in the dimensionally-reduced space.

(2) Graph creation: SciPy (*Virtanen et al., 2020*) for evaluating the distances and similarities between the rows of the dataset and NetworkX (*Hagberg, Swart & Schult, 2008*) for handling the graph.

(3) Graph analysis: Matplotlib (*Hunter, 2007*) for the visualization and NetworkX (*Hagberg, Swart & Schult, 2008*) for the centrality measurements and community detection algorithms.

All these dependencies can be installed *via* the `requirement.txt` in our GitHub repository or automatically with `pip`.

### Software functionalities

In this section we explain the package with its off-the-shelf features.

#### Configuration file

TaGra's flexibility is enhanced by its support for a JSON configuration file, allowing users to specify preprocessing, graph creation, and analysis settings. This ensures reproducibility and ease of use. Users can customize preprocessing options, such as handling missing values, encoding categorical variables, scaling numeric features, and applying manifold learning. The graph creation parameters are also flexible, supporting various methods with

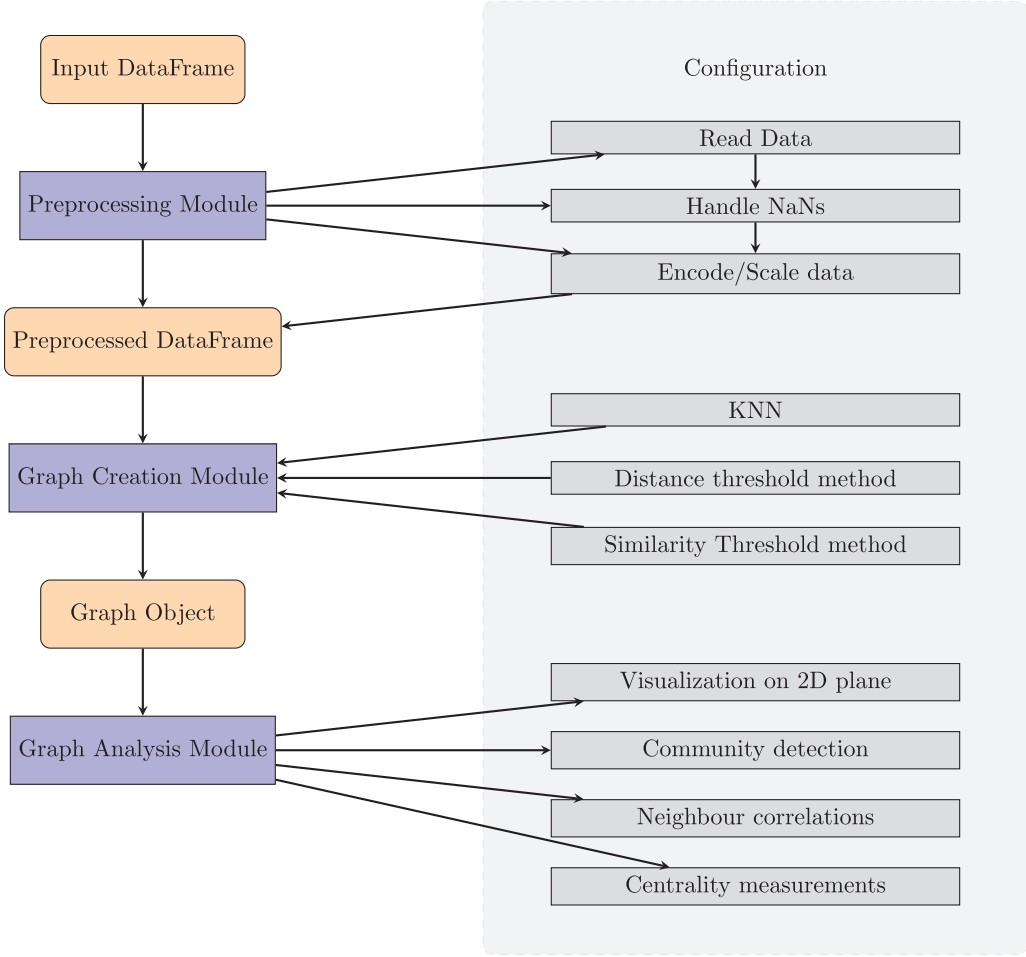

**Figure 1 Diagram of TaGra's operation.** The three modules, with their options, can be used consecutively to obtain a graph from a set of tabular data or individually.

customizable settings to adapt to different analysis needs. Through this subsection, we provide a description of both the submodules and their options that can be specified in the configuration file. Even though it is possible for a user to specify only the dataframe path and the name of the target variable, it is possible to customise the processing and analysis by changing the options in a JSON file. All options are listed in Table 1.

## Data preprocessing

This module automates handling missing values, encoding categorical variables, and scaling numeric features.

Specify the input dataframe in the configuration file under `input_dataframe` (supported extensions: CSV, Microsoft Excel, Pickle, JSON, Parquet, HDF5). Columns can be designated as numeric or categorical *via* `numeric_columns` and `categorical_columns`. Categorical columns are encoded (One-hot or Label encoding) and numeric columns are scaled (MinMax or Standard Scaler).

**Table 1 Complete configuration flags of TaGra.**

| Option | Description | Default value |
|---|---|---|
| input_dataframe | Path to the input DataFrame. | *mandatory* |
| output_directory | Path to the folder where the results will be collected. | results/ |
| preprocessed_filename | Filename of the preprocessed DataFrame. | None |
| graph_filename | Filename of the graph file. | None |
| numeric_columns | List of columns to be treated as numeric. | None |
| categorical_columns | List of columns to be treated as categorical. | [ ] |
| target_columns | Column to be used as the target variable for coloring the graph and neighborhood statistics. | [ ] |
| ignore_columns | List of columns to ignore during preprocessing. | [ ] |
| unknown_column_action | Action to take on columns not specified as numeric, categorical, or ignored. | "infer" |
| numeric_threshold | Threshold for determining if a column is numeric. | 0.05 |
| numeric_scaling | Method for scaling numeric columns. | "standard" |
| categorical_encoding | Method for encoding categorical columns. | "one-hot" |
| nan_action | Action to take on NaN values. | "infer" |
| nan_threshold | Threshold for dropping columns based on NaN ratio. | 0.5 |
| verbose | Flag for detailed output. | True |
| overwrite | Overwrite or not the previous output. | False |
| manifold_method | Method for manifold learning on numeric columns. | None |
| manifold_dimension | Number of dimensions for manifold learning output. | None |
| method | Method for creating the graph. | "knn" |
| k | Number of neighbors for KNN graph creation. | 5 |
| distance_threshold | Threshold for distance-based graph creation. | None |
| similarity_threshold | Threshold for similarity-based graph creation. | None |
| neigh_prob_path | Filename for neighborhood probability statistics. | "neigh_prob.txt" |
| degree_distribution_filename | Filename for the degree distribution plot. | "degree.png" |
| community_filename | Filename for the community composition histogram. | "communities.png" |
| graph_visualization_filename | Filename for the graph visualization. | "graph.png" |
| prob_heatmap_filename | Filename for the neighborhood probability heatmap. | "neigh_prob_heatmap.png" |
| network_metrics_filename | Filename for the other network metrics. | None (will be displayed in terminal) |

Use `target_columns` to specify the target variable for class differentiation, and `ignore_columns` to exclude columns from processing. The `unknown_column_action` flag (infer or ignore) handles unspecified columns, using `numeric_threshold` to determine numeric classification.

Manage missing data with `nan_action` (drop row, drop col, or infer), and set `nan_threshold` for column removal.

Optionally, apply manifold learning Isomap (*Tenenbaum, Silva & Langford, 2000*), *t*-distributed Stochastic Neighbor Embedding (*t*-SNE) (*Van der Maaten & Hinton, 2008*) and Uniform Manifold Approximation and Projection (UMAP) (*McInnes, Healy & Melville, 2018*; *Healy & McInnes, 2024*) to numeric data using the `manifold_method`

flag. The preprocessed file is saved using `preprocessed_filename` and `output_directory`.

### Graph creation

Once the data is preprocessed, TaGra's graph creation module transforms the cleaned data into a graph. Each node corresponds to a row in the dataset and contains the complete feature vector as its attributes, while the edges between the nodes represent a similarity relationship between the nodes. Users can choose from three methods to set these similarity: K-nearest neighbors (KNN, *Zhang et al., 2017*), Distance Threshold (Radius Graph, *Carneiro & Zhao, 2018*), and Similarity Graph (*Singhal, 2001*). The graph creation mode can be set with `method` (either `"knn"`, `"distance"` or `"similarity"`). The KNN method constructs a graph by connecting each node to its $k$ nearest neighbors based on Euclidean distance. The value of $k$ can be set with `"k"`. The Distance Threshold method creates edges between nodes if their Euclidean distance is below a specified threshold (`"distance_threshold"`), which is ideal for identifying closely related data points. The Similarity Graph method adds edges between nodes if the cosine similarity of between two nodes is greater than `"similarity_threshold"`.

### Graph analysis

The graph analysis module facilitates comprehensive examination of network structures generated in the previous processing step. While designed to work seamlessly with graphs produced by the preceding modules, this component also supports independent analysis of externally generated networks, provided they conform to the GraphML format specification. While several established tools exist for network visualization and analysis (for example, NetworkX (*Hagberg, Swart & Schult, 2008*), GraVis (*Haas, 2022*), NetworKit (*Staudt, Sazonovs & Meyerhenke, 2016*), Gephi (*Bastian, Heymann & Jacomy, 2009*), and Cytoscape (*Shannon et al., 2003*)), our approach offers a unique workflow that bridges manifold learning with network analytics.

First, the graph is visualised and saved in the output directory under the name specified with `"graph_visualization_filename"`. For node positioning, we uniquely leverage the coordinates obtained from manifold learning in the previous step, creating a natural connection between dimensionality reduction and network visualization. When these are not available, we default to the Fruchterman-Reingold force-directed algorithm (*Fruchterman & Reingold, 1991*) with the NetworkX implementation. If the target variable is specified, each node (the rows of the previous step) are colored according to the value of that attribute, and we perform a distinctive neighborhood analysis that evaluates the probability of extracting a neighbor with label $j$ given that the selected node has attribute $i$. The output of this analysis is both printed on the screen and saved in `"neigh_prob_path"` in txt file and `"prob_heatmap_filename"` as an heatmap in which the cell $(i, j)$ tells the aforementioned probability. This neighborhood probability heatmap provides a unique analytical perspective for assessing network structure. Intuitively, diagonal elements approaching 1 indicate strong class separation,

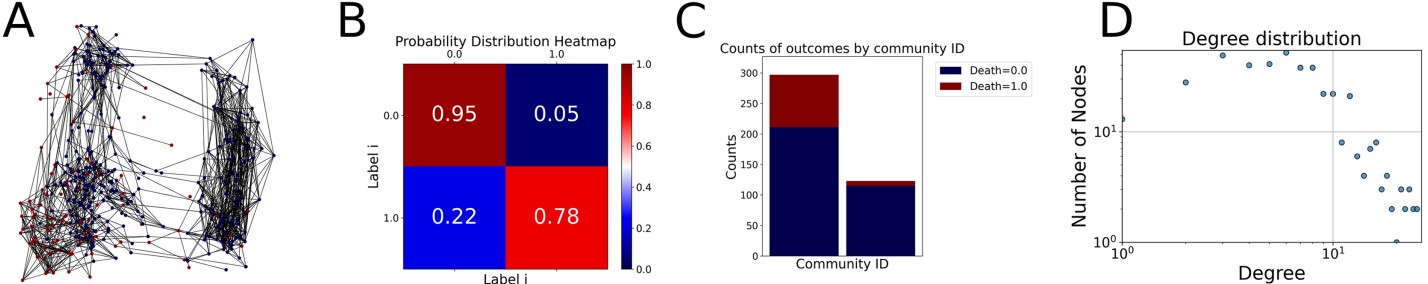

**Figure 2 Outputs of the graph analysis module generated from the heart failure EHRs dataset.** (A) Graph visualization illustrates connectivity patterns between data points based on similarity relationships. Isolated nodes (that are, nodes with no edges) represent potential outliers, as they lack connections to other nodes. (B) Neighbor probability heatmap indicates the likelihood of each data point being connected to its neighbors, offering insights into local density and neighborhood structure. (C) Community composition histogram reveals clusters within the graph, facilitating the interpretation of group separability and data organization. (D) Degree distribution plot displays the network connectivity pattern, identifying potential outliers (nodes with few connections) and central nodes (nodes with many connections). These outputs were generated using the Similarity Graph method. For further details, refer to the Graph Creation paragraph. EHR, electronic health records; HF, heart failure.

as nodes predominantly connect with same-class neighbors. Conversely, weaker diagonals suggest poorer separation, offering researchers a quantitative measure of homophily or assortative mixing that is not directly available in standard network analysis tools.

Then the degree distribution of the node is evaluated and saved under `"degree_distribution_filename"` if specified. Following this, the Girvan Newman algorithm (*Girvan & Newman, 2002*) search the communities and their composition are saved in `"community_filename"`. The resulting visualization displays community ID on the x-axis and cardinality on the y-axis. When a target variable is specified, our tool provides an enhanced community analysis by coloring each bar according to the class distribution within that community, offering insights into how class labels are distributed across the detected network structures—a feature particularly valuable for analyzing the relationship between community structure and node attributes. In Fig. 2 we summarized the visuaizations with a brief explaination. TaGra's graph analysis module extends beyond basic visualization capabilities to provide a comprehensive statistical evaluation of graph structure and quality. The module calculates several categories of metrics that quantify different aspects of the network topology: *graph density*, *average clustering coefficient*, *number of connected components*, and *size of the largest component* (for definitions of these metrics, see *Newman, 2018*).

Furthermore, TaGra evaluates class separation metrics to assess how well the graph preserves relationships between nodes with different target attributes. Specifically, we compute: (1) chi-square tests on the contingency table of neighbor relationships to detect non-random connectivity patterns; (2) a *homophily score*, defined as the frequency of same-class connections relative to total connections; and (3) statistical significance assessments through permutation tests and z-scores to determine whether the observed class separation exceeds what would be expected by random chance.

### Quickstart

You can run these three steps consecutively with the simple bash command `python3 go. py -d path/to/dataframe -a target_column_name`.

In the bash shell the path to the saved file is displayed as well as the elapsed time and the neighbor probabilities (if the target variable is specified).

## RESULTS

In this section, we apply the methodologies described in the previous section and proceed with the analysis of two datasets derived from electronic health records (EHRs), one concerning cases of comorbidity of depression and heart failure (*Jani et al., 2016*) and one concerning cases of diabetes type one (*Smith et al., 1988*). For both, we will quickly describe the content and report the visualisation of the graphs obtained from the dataset, complete with the analysis of the topological structures of the graphs themselves.

### Depression and heart failure EHRs dataset

We consider a comorbid Depression and Heart Failure (HF) dataset of electronic health records (EHRs) (*Jani et al., 2016*), a high-dimensional dataset featuring data from 425 patients, with 10 features (excluding the *id* of the patient). Here, we demonstrate how the output graph properties change when specifying one of three different graph construction methods and the underlying statistics. The layout of the graph (that is, the position of the nodes) is evaluated by reducing the space using the Isomap method for each method. For the preprocessing, we specified only the target column and a single column to be ignored, namely the *id* which should not be taken in consideration when evaluating the distance or similarity relations between nodes. Tables 2 and 3 summarize the experiments, coloring nodes red (target = 0) or blue (target = 1).

With the distance threshold method, central nodes indicate typical data points, while fewer connections suggest outliers. Neighbor probabilities show a 53% chance of finding a class 0 neighbor given class 1, and community composition reveals a large central community with smaller ones around it, suggesting this method excels at identifying typical nodes and outliers but not class separation: as we mentioned in the last section, if a good separation were achieved by this method, then the neighbors of a node with a certain class should have the same class as the node under consideration, and the elements off the diagonal of the matrix should be around 0%.

The K-nearest neighbors method produces a graph with fewer connections and no evident central nodes, resulting in fewer communities and some separability between neighbors, as seen in the community histogram and neighbor probability heatmap.

The similarity threshold method results in two communities and central nodes, with an improved neighbor matrix and better neighbor separation compared to K-nearest neighbors.

Looking at the quantitative metrics for the HF EHRs dataset across the three graph representation methods in Table 3, several important patterns emerge: similarity method demonstrates the strongest class separation with the highest homophily score (0.87) and homophily Z-score (29.75), indicating that nodes predominantly connect with others of

**Table 2** Heart failure EHRs dataset—comparison of three different graph representation methods.

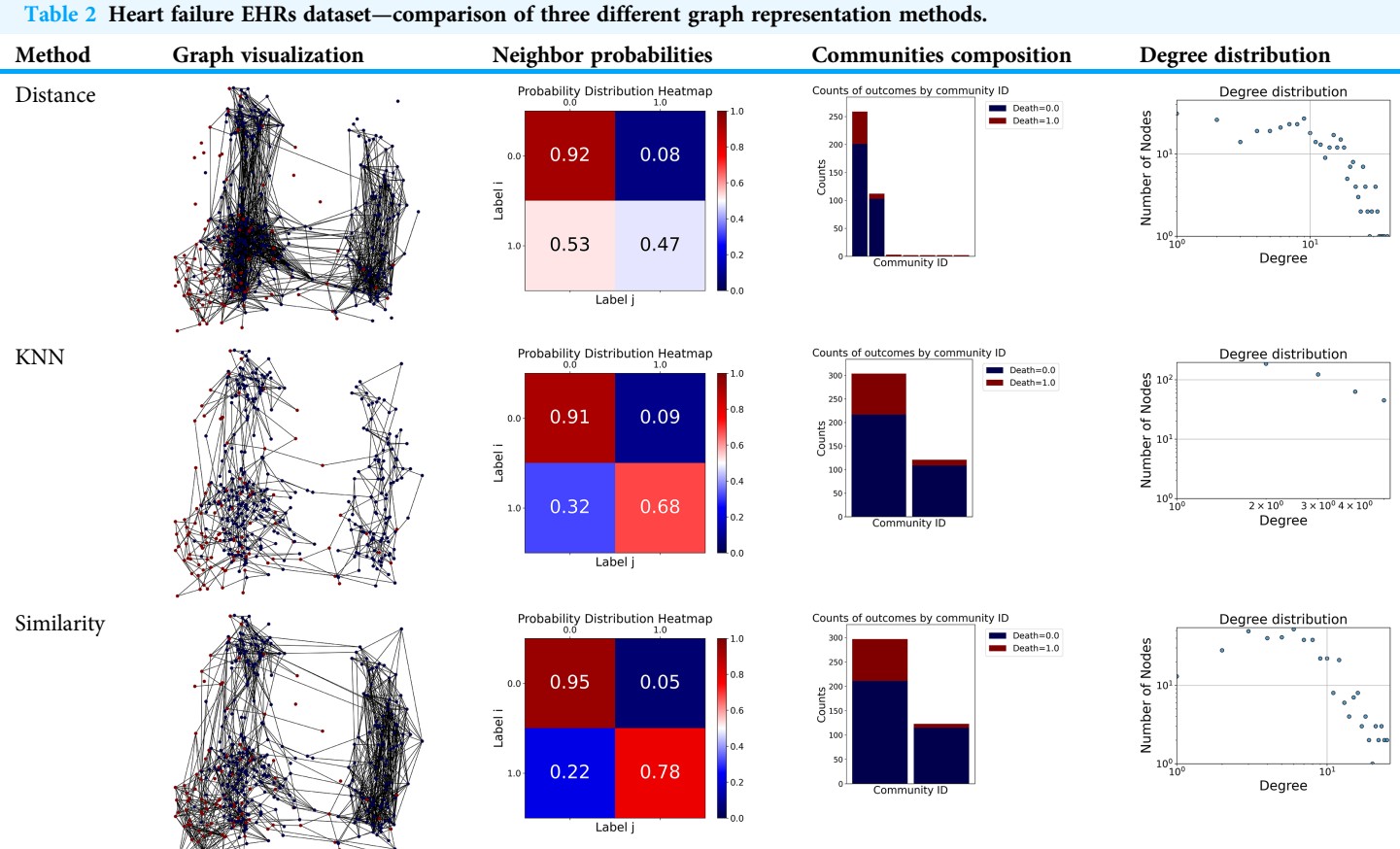

| Method | Graph visualization | Neighbor probabilities | Communities composition | Degree distribution |
|---|---|---|---|---|
| Distance | | | | |
| KNN | | | | |
| Similarity | | | | |

**Note:**
Each row indicates a methodology to generate the graph and each column contains one of the output results of TaGra. We reported a detailed explanation of these output results in Fig. 2. EHRs, Electronic health records; HF, Heart failure.

**Table 3** Heart failure EHRs dataset—comparison of three different graph representation methods.

| Metric | Distance threshold | K-nearest neighbors | Similarity |
|---|---|---|---|
| Graph density | $2.29 \times 10^{-2}$ | $0.71 \times 10^{-2}$ | $1.70 \times 10^{-2}$ |
| Average clustering coefficient | 0.34 | 0.25 | 0.35 |
| Connected components | 49 | 1 | 6 |
| Largest component size (%) | 87.3% | 100.0% | 98.8% |
| Assortativity coefficient | 0.31 | −0.06 | 0.60 |
| Community count | 50 | 2 | 7 |
| Modularity score | 0.39 | 0.39 | 0.48 |
| Homophily score | 0.70 | 0.80 | 0.87 |
| Chi-square $p$-value | 0.00 | 0.00 | 0.000 |
| Homophily Z-score | 18.1 | 14.3 | 29.75 |

**Note:**
Quantitative metrics for each graph construction approach. Each metric is defined in Section Methods.

the same class far beyond what would be expected by random chance. This is further supported by its high assortativity coefficient (0.60), showing that similar nodes tend to connect to each other. While the K-nearest neighbors approach produces a perfectly connected graph (single component with 100% coverage), it shows lower clustering (0.25) and negative assortativity (−0.06), suggesting less meaningful local neighborhood structures. Nevertheless, it maintains good homophily (0.80), indicating decent class separation. The Distance Threshold method yields the most fragmented graph (49 connected components), which explains its higher community count (50). Despite having the highest density ($2.29 \times 10^{-2}$), its class separation metrics are the weakest among the three approaches, though still statistically significant (homophily Z-score of 18.1). All three methods show statistically significant non-random connectivity patterns ($p$-value near 0), but the Similarity method achieves the best balance between meaningful community structure (modularity score of 0.48) and class separation. This quantitative analysis confirms our visual observations and demonstrates that the Similarity method most effectively captures the inherent structure of the HF dataset while preserving meaningful class relationships.

TaGra demonstrates exceptional capabilities in data analysis through its robust graph-based visualization framework. The graph visualization component effectively illustrates data connectivity patterns, emphasizing the detection of potential outliers as isolated nodes—those without edge connections to other vertices in the graph. The neighbor probability heatmap provides quantitative insights into local density structures and neighborhood relationships, where optimal separation is achieved when nodes with target label 0 connect exclusively with other label 0 nodes, and similarly for label 1 nodes. While perfect separation is ideal, the presence of strong diagonal values in the probability heatmap indicates effective data partitioning and minimal overlap between classes.

In the dimensionally reduced space, the graph visualization highlights two primary data clusters with sparse inter-cluster connections, suggesting distinct separability. The community composition histogram further characterizes these clusters, revealing that the minority cluster is predominantly composed of target 1 data points, reflecting class imbalance. The degree distribution analysis completes the topological assessment by identifying peripheral nodes (potential outliers with minimal connections) and hub nodes (central vertices with numerous connections). The heavy tail observed in the double-log plot of degree distribution reflects a hierarchical structure in the dataset, where a few highly connected nodes (typical data points) coexist with many low-degree nodes, reinforcing the presence of varying data importance and prominence within the graph.

These analytical outputs, generated through the Similarity Graph method, underscore both the necessity of addressing outliers for enhanced data quality and the potential benefits of employing non-linear methodologies to capture complex underlying patterns in the data structure.

## Diabetes type one EHRs dataset

Let us consider the type one diabetes dataset (*Smith et al., 1988*), which reports data from 768 patients, each with nine features. Here, we demonstrate how the output graph

**Table 4 Diabetes type one EHRs dataset—comparison of three different graph representation methods.**

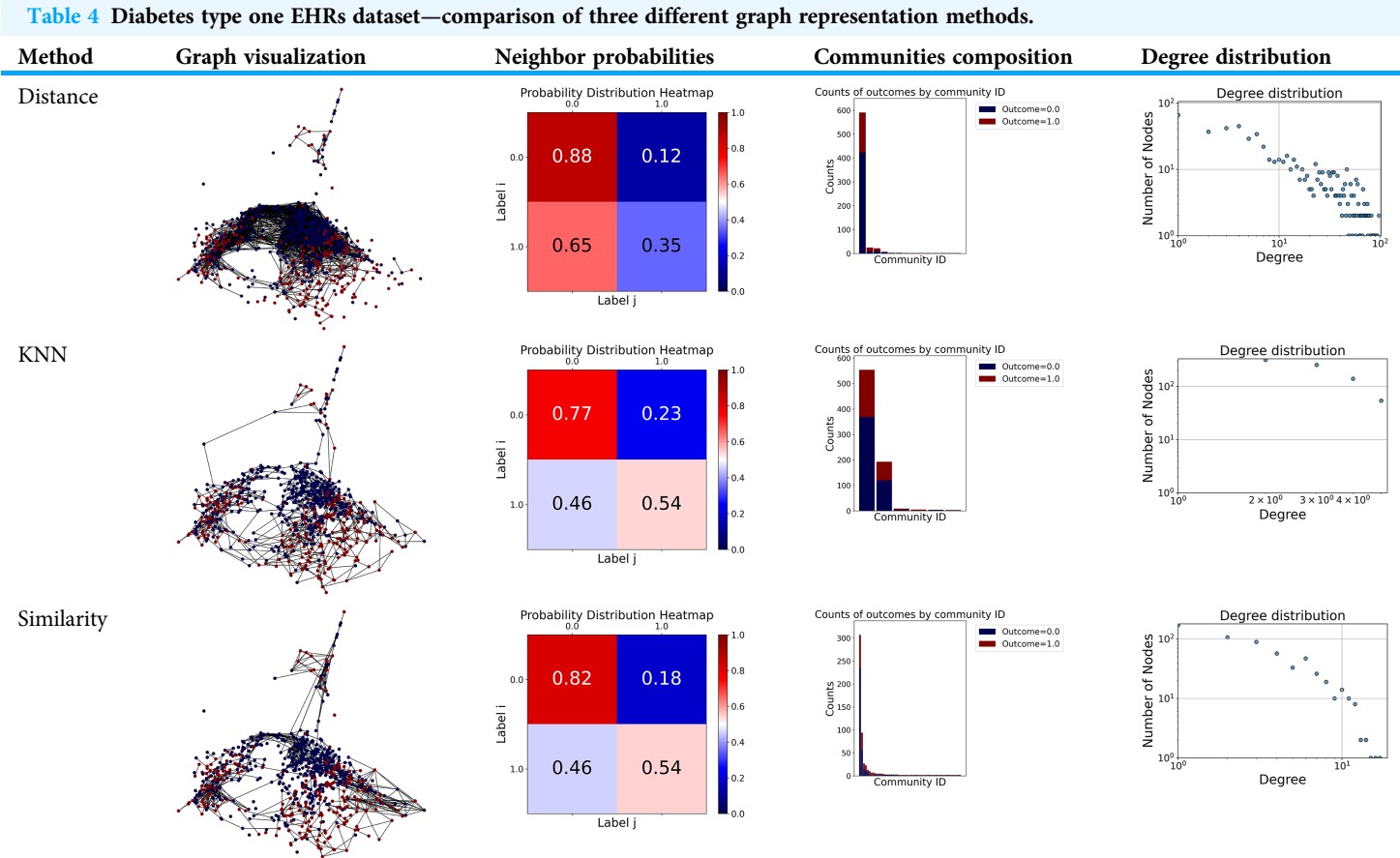

| Method | Graph visualization | Neighbor probabilities | Communities composition | Degree distribution |
|---|---|---|---|---|
| Distance | | | | |
| KNN | | | | |
| Similarity | | | | |

**Note:**
Each row indicates a methodology to generate the graph and each column contains one of the output results of TaGra. We reported a detailed explanation of these output results in Fig. 2.

**Table 5 Diabetes type one EHRs dataset—comparison of three different graph representation methods.**

| Metric | Distance threshold | K-nearest neighbors | Similarity |
|---|---|---|---|
| Graph density | $2.35 \times 10^{-2}$ | $0.38 \times 10^{-2}$ | $0.37 \times 10^{-2}$ |
| Average clustering coefficient | 0.38 | 0.24 | 0.23 |
| Connected components | 114 | 5 | 222 |
| Largest component size (%) | 80.2% | 97.3% | 52.2% |
| Assortativity coefficient | 0.52 | −0.04 | 0.58 |
| Community count | 115 | 6 | 223 |
| Modularity score | 0.03 | 0.40 | 0.58 |
| Homophily score | 0.61 | 0.65 | 0.68 |
| Chi-square $P$-value | 0.00 | 0.00 | 0.000 |
| Homophily Z-score | 18.8 | 10.4 | 14.15 |

**Note:**
Quantitative metrics for each graph construction approach. Each metric is defined in Section Methods.

properties change when specifying one of three different graph construction methods and the underlying statistics. The layout of the graph (that is, the position of the nodes) is evaluated by reducing the space using the Isomap method for each method. For the preprocessing, we specified only the target column and a single column to be ignored, namely the *id* which should not be taken in consideration when evaluating the distance or similarity relations between nodes. Tables 4 and 5 reports the three different experiments. The nodes are colored blue, corresponding to patient who has not been diagnosed with diabetes mellitus (outcome = 0) or red, corresponding patient who has been diagnosed with diabetes mellitus (outcome = 1).

In the graph obtained with the distance threshold method, we observe nodes that are more central than others (see graph visualization and degree distribution). This information helps us infer the presence of outliers (which have fewer connections) and typical data points (which have a larger number of connections). Additionally, this reasoning can be extended to multiple data items by examining nodes that are strongly connected but cluster in small communities. By combining information on communities and individual node connections, it is possible to identify outliers among multiple data items. Additionally, the neighbor probabilities indicate a 65% chance of finding a neighbor of target class 0 given that a node has class 1. The community composition reveals a large central community with many smaller surrounding communities, each with a similar proportion of nodes of class 0 and 1. These insights suggest that while this method is not effective at class separation, it excels in identifying typical nodes and outliers.

With K-nearest neighbors, a graph with fewer connections and no central nodes is created. Consequently, we have fewer communities, but we observe a certain degree of separability between neighbors, as observed from the community histogram and neighbor probability heatmap.

Finally, in the graph obtained with the similarity threshold method, the number of communities is higher, but we notice both the presence of central nodes (from the degree distribution) and, compared to K-nearest neighbors, an improved neighbor matrix, resulting in better neighbor separation.

The quantitative metrics for the Diabetes EHRs dataset reveal distinct characteristics across the three graph construction methods: similarity method shows the highest modularity score (0.58) and strong assortativity (0.58), indicating well-defined community structures and preferential connections between similar nodes. However, it produces a highly fragmented graph with 222 connected components and only 52.2% of nodes in the largest component. This fragmentation suggests that while local relationships are preserved with good homophily (0.68), the global structure is less cohesive compared to other methods.

The K-nearest neighbors approach creates a more connected graph with only five components and 97.3% of nodes in the largest component. It achieves moderate modularity (0.40) but shows slight disassortativity (−0.04), indicating connections between dissimilar nodes. The homophily score (0.65) demonstrates reasonable class separation, though with a lower Z-score (10.4) than other methods.

The Distance Threshold method generates a dense graph ($2.35 \times 10^{-2}$) with high clustering coefficient (0.38) and positive assortativity (0.52). Despite having 114 components, it maintains good coverage with 80.2% of nodes in the largest component. Its relatively low modularity score (0.03) suggests less distinct community structures, though it achieves solid homophily metrics (0.61 score, 18.8 Z-score).

All three methods show statistically significant class separation (*p*-values of 0.00), but each offers different trade-offs between connectivity, community structure, and class separation. For the Diabetes dataset, the choice of method depends on analytical priorities: K-nearest neighbors provides the best balance between connectivity and community structure, while Similarity offers stronger local class separation but at the cost of global connectivity.

This analysis highlights how different graph construction approaches can reveal distinct aspects of the underlying data structure, emphasizing the value of TaGra's multiple graph generation methods when analyzing complex biomedical datasets.

We obtained a data visualization in which the similarity relations are marked by the presence of an edge between two nodes (the individual data items). It is possible to detect potential outliers (in the distance and similarity threshold methods, there are several isolated nodes), and we can see that given the different methods, it is challenging to separate the data both locally around a node (neighbor probabilities) and at the cluster level (community composition). This analysis suggests that a dataset like this should be cleaned of its outliers and treated with non-linear methods.

### Reproducing the results

To comply with the reproducibility principles of open science (*Sandve et al., 2013*; *Iqbal et al., 2016*), we report here the instructions to reproduce the results described in this study from any personal computer having Python installed and an internet connection. After downloading the repository from https://github.com/davidetorre92/TaGra and installing the required dependencies, the results presented in this analysis can be reproduced using the following command:

```
python3 go.py -c examples/article/dataset/{method}.json
```

Where `{dataset}` can be specified as either `diabetes` or `hf`, and `{method}` can be set to `distance`, `knn`, or `similarity`. The execution generates output files in the directory `dataset/article/output/{dataset}` containing all visualizations and analyses presented above.

## COMPARISON WITH EXISTING GRAPH-GENERATION TOOLS

Our package fills a niche between raw graph libraries (for example, igraph (*Csardi & Nepusz, 2006*; *Csárdi et al., 2025*)) and visualization tools by providing an attribute-focused analysis pipeline with automated reporting. Unlike NetworkX's (*Hagberg, Swart & Schult, 2008*) general-purpose functions or igraph's performance-oriented approach, TaGra specializes in quantifying and visualizing node attribute relationships through probability metrics and heatmaps. While lacking the scalability of graph-tool (*Peixoto, 2014*) or the

**Table 6 Functionality comparison between TaGra and related tools: NetworkX (*Hagberg, Swart & Schult, 2008*), GraVis (*Haas, 2022*), NetworKit (*Staudt, Sazonovs & Meyerhenke, 2016*), Gephi (*Bastian, Heymann & Jacomy, 2009*).**

| Functionality | TaGra | NetworkX | Gephi | igraph | NetworKit |
|---|---|---|---|---|---|
| Automated preprocessing of tabular data with categorical variables handling | ✓ | × | × | × | × |
| Multiple graph creation methods | ✓ | ✓ | × | ✓ | ✓ |
| Manifold learning integration | ✓ | × | × | × | × |
| Neighborhood probability analysis | ✓ | × | × | × | × |
| Community detection | ✓ | ✓ | ✓ | ✓ | ✓ |
| Target variable-based visualization | ✓ | ○ | ○ | ○ | ○ |
| Single command end-to-end analysis | ✓ | × | × | × | × |
| Interactive visualization | × | × | ✓ | × | × |
| Large-scale graph processing | ○ | ○ | ✓ | ✓ | ✓ |

**Note:**
✓, Native support; ○, Partial support (requires additional coding); ×, Not supported.

interactivity of Gephi (*Bastian, Heymann & Jacomy, 2009*), our TaGra library offers unique value in automated multi-output analysis (text reports + plots) for tabular-data-derived graphs.

Regarding specialized network visualization libraries like GraVis (*Haas, 2022*) and NetworKit (*Staudt, Sazonovs & Meyerhenke, 2016*), our tool differentiates itself through the integration with manifold learning and its focus on neighbourhood analysis for attributed graphs. While GraVis and NetworKit provide excellent general-purpose network visualization, they do not offer the same tailored approach to analyzing the relationship between node attributes and network structure through probability-based metrics.

Table 6 provides a functionality comparison between TaGra and related tools, highlighting our package's unique capabilities in preprocessing tabular data, handling categorical variables, and performing neighborhood probability analysis—features not natively supported by existing graph libraries.

We developed our TaGra software independently and concurrently with some of these specialized visualization libraries, which speaks to the growing recognition of the need for attribute-focused network analysis tools. Looking forward, we plan to implement export capabilities to established visualization platforms like Gephi (*Bastian, Heymann & Jacomy, 2009*) and Cytoscape (*Shannon et al., 2003*), allowing users to further explore the generated networks with these interactive tools. We also aim to incorporate selected functionality from NetworKit to improve performance for larger graphs while maintaining our unique analytical perspective. This makes TaGra particularly suited for exploratory data analysis of medium-sized attributed graphs, complementing rather than replacing existing libraries.

## DISCUSSION AND CONCLUSIONS

We have presented software whose aim is to provide a basic tool for pre-processing and visualisation of data in many dimensions. It serves as a tool for rapid preliminary analysis

of datasets characterized by a large number of features. Through the creation and visualization of relationships between points in a dataset, it is possible to reduce the dimensionality while preserving the proximity or similarity information of each point with respect to its neighborhood, thus retaining more information than the simple dimensionalityreduction (*Geng, Zhan & Zhou, 2005*; *Jia et al., 2022*).

In addition, this approach is informative as it helps detect data items that are more typical than others. In the last section, we showed that some data items are particularly central to the dataset, indicating that their similarity relationships are consistent at many entry points. Conversely, other data items have fewer connections with the rest of the dataset. This allows us to identify data items with more similarity relationships and outliers. The identification and classification of data entries into these two categories are essential for data analysis (*Ramaswamy, Rastogi & Shim, 2000*; *Sullivan, Warkentin & Wallace, 2021*; *Alghushairy et al., 2020*).

Finally, the community detection algorithm and the neighbor probability distribution aid the user in understanding if the dataset is separated in two ways: first by determining if the separation occurs locally at the level of each node, and then if the separation happens on a larger scale, *i.e.*, at the community level. Understanding the separations within a dataset is crucial for determining the appropriate techniques and methods to use for further analysis and classification tasks (*Sedlmair et al., 2012*).

To summarize the contribution of the software we presented here, it is a way to visualize high-dimensional data while preserving the proximity and similarity relationships between data points. This approach not only facilitates the detection of typical and atypical data items but also aids in understanding local and global separations within the dataset. By providing tools for data preprocessing, graph creation, and community analysis, our software enables users to effectively explore and analyze complex datasets, thus enhancing their ability to make informed decisions about subsequent analytical and classification tasks.

Through its integration with widely-used Python libraries and support for various data formats, TaGra offers flexibility and ease of use, aiding more informed and efficient data analysis.

## Limitations

A notable limitation of TaGra lies in its dependency on the number of edges in the constructed graph, which heavily influences the effectiveness of the community detection module. On the other hand, sparse graphs with fewer edges may fail to capture meaningful cluster structures, leading to fragmented or incomplete communities. Additionally, in the current implementation, distance and similarity matrices used for graph generation rely on predefined metrics, which may not fully adapt to complex data structures. Incorporating advanced methods such as those described in TMAP (*Probst & Reymond, 2019*), which dynamically adjust similarity measures for large-scale datasets, could enhance the robustness and scalability of graph construction and improve downstream analyses. In the Supplemental Information we discuss an additional limitation related to the application of

Tagra to an EHRs dataset containing a small number of entries (*Takashi et al., 2019*; *Cerono & Chicco, 2024*). The tool faces scalability challenges with very large datasets, particularly in high-dimensional spaces where distance-based methods often perform poorly. Current graph construction methods create unweighted edges, which may not adequately capture relationship strengths between nodes. Graph quality remains sensitive to user-specified parameters (like $k$ in KNN, or threshold values), requiring careful tuning for optimal results.

For input data, TaGra can be sensitive to noise—outliers or erroneous values may significantly impact the resulting graph structure and subsequent analysis. While the preprocessing module provides options for handling missing values and scaling, the quality of the final visualization directly depends on data cleanliness. This is particularly relevant for EHRs datasets, which often contain measurement errors or inconsistently recorded values. From an integration perspective, TaGra's current API structure provides basic export capabilities to formats compatible with tools like Gephi and Cytoscape, but lacks direct integration bridges to these platforms. The extensibility of the codebase for implementing custom graph construction methods or specialized neighborhood analyses requires improvement in future versions. A limitation of this is study is that we showed the effectiveness of TaGra only on datasets of medical records, and we did not show its efficacy on other data types. We selected EHRs data because the data of this particular type are not collected for scientific purposes, and consists of data variables of different types (numeric, ordinal, categorical, binary, *etc*.), making their scientific analysis particularly challenging. Due to the ease with which medical records data are collected (usually a blood test is sufficient to have tens of clinical factors), computational analyses on them can lead to impactful and useful scientific discoveries, at limited cost in terms of resources and money. However, the use of only two EHRs datasets, even if they pertain to two different diseases, diminishes the generalizability of our findings. In the future, we plan to apply TaGra to other types of biomedical data as well.

## Future works

To further enhance TaGra's capabilities, future development will focus on integrating advanced graph analysis techniques, supporting real-time data processing, and expanding its functionality for other data domains, such as social network analysis (*Scott, 2000*). Social networks often consist of highly dynamic and interconnected structures that evolve over time (*Braha & Bar-Yam, 2009*). By incorporating support for dynamic graphs, TaGra could analyze time-evolving networks, capturing the changes in connectivity patterns and community structures as they occur. This functionality would be particularly valuable in applications such as tracking information flow, detecting emerging trends, or monitoring influence propagation in social media platforms. Additionally, advanced graph metrics such as betweenness centrality, closeness centrality, and graph embeddings could provide deeper insights into the hierarchical roles of nodes and uncover influential entities or hidden relationships within social networks.

Computational genomics and proteomics in general can be another scientific area where TaGra can be effectively applied. Networks of protein-protein interactions (PPIs), in particular, seem specifically suitable for our method: applied to this kind of network, TaGra would produce an informative representation of the protein-protein interactions, and potentially infer new knowledge about the proteome. Protein-protein interaction networks can be found in open public databases such as KEGG (*Kanehisa et al., 2017*), Reactome (*Croft et al., 2010*), or STRING (*Szklarczyk et al., 2024*).

Additionally, we envision effective applications of our software package of pharmacological data, too. Network-based approaches in this field, in fact, can unveil new relationships between chemical compounds and drug components (*Boezio et al., 2017*; *Moon & Rho, 2025*). We also plan to include additional visualization interactive features (*Brown & Chicco, 2024*), and to propose the inclusion of TaGra into open biomedical informatics platforms such as Galaxy (*Jalili et al., 2020*), Bioconda (*Grüning et al., 2018*), or Bioconductor (*Amezquita et al., 2020*).

## LIST OF ABBREVIATIONS

| | |
|---|---|
| **API** | application programming interface |
| **CC** | Creative Commons |
| **CSV** | comma-separated values |
| **HF** | Heart failure |
| **EHRs** | electronic health records |
| **HDF5** | Hierarchical Data Format |
| **JSON** | JavaScript Object Notation |
| **k-NN** | KNN: $k$-nearest neighbors |
| **KEGG** | Kyoto Encyclopedia of Genes and Genomes |
| **PPIs** | protein-protein interaction networks |
| **PyPI** | Python Package Index |
| $t$-**SNE** | t-distributed stochastic neighbor embedding |
| **PHATE** | Potential of Heat-diffusion for Affinity-based Transition Embedding |
| **TMAP** | Tree MAP |
| **UMAP** | Uniform Manifold Approximation and Projection for Dimension Reduction |

## ACKNOWLEDGEMENTS

Davide Torre is a PhD student enrolled in the National PhD in Artificial Intelligence for Health and Life Sciences, XXXVII cycle (Università Campus Bio-Medico di Roma). We acknowledge and credit the use of large language models (LLMs), specifically Claude by Anthropic, in this work. Claude assisted with validating our manuscript's formal structure, improving linguistic clarity, and refining syntactical elements. All AI-generated content underwent thorough human review and validation to ensure accuracy and alignment with our research objectives. The scientific analysis, methodology development, and interpretation of results remain entirely the work of the human authors.

## Funding

This study was funded by the Italian Ministero Italiano delle Imprese e del Made in Italy under the Digital Intervention in Psychiatric and Psychologist Services (DIPPS) (project code F/310240/01-04/X56) programme within the framework 'Innovation Agreements' (Accordi per l'Innovazione) and by Ministero dell'Università e della Ricerca of Italy under the 'Dipartimenti di Eccellenza 2023–2027' ReGAInS grant assigned to Dipartimento di Informatica Sistemistica e Comunicazione at Università di Milano-Bicocca. There was no additional external funding received for this study. The funders had no role in study design, data collection and analysis, decision to publish, or preparation of the manuscript.

## Grant Disclosures

The following grant information was disclosed by the authors:
Italian Ministero Italiano delle Imprese e del Made in Italy.
Digital Intervention in Psychiatric and Psychologist Services (DIPPS): F/310240/01-04/X56.
Ministero dell'Università e della Ricerca of Italy.
ReGAInS grant assigned to Dipartimento di Informatica Sistemistica e Comunicazione at Università di Milano-Bicocca.

## Competing Interests

Davide Chicco is an Academic Editor for PeerJ.

## Author Contributions

- Davide Torre conceived and designed the experiments, performed the experiments, analyzed the data, performed the computation work, prepared figures and/or tables, authored or reviewed drafts of the article, and approved the final draft.
- Davide Chicco conceived and designed the experiments, prepared figures and/or tables, authored or reviewed drafts of the article, and approved the final draft.

## Data Availability

The TaGra software package is available at PyPI, GitHub and Zenodo:

- PyPI https://pypi.org/project/TaGra, MIT License;
- https://github.com/davidetorre92/TaGra;
- Davide Torre. (2024). davidetorre92/TaGra: TaGra (0.2.3). Zenodo. https://doi.org/10.5281/zenodo.13142796.

The Heart-Failure Dataset is available at figshare: Dinesh Jani, Bhautesh; S. Mair, Frances; Roger, Véronique L.; Weston, Susan A.; Jiang, Ruoxiang; M. Chamberlain, Alanna (2016). Comorbid Depression and Heart Failure: A Community Cohort Study. PLOS ONE. Dataset. https://doi.org/10.1371/journal.pone.0158570.

The Diabetes Dataset is available at Kaggle: https://www.kaggle.com/datasets/uciml/pima-indians-diabetes-database, CC0. It was originally available at https://archive.ics.uci.edu/dataset/34/diabetes.

## Supplemental Information

Supplemental information for this article can be found online at http://dx.doi.org/10.7717/peerj-cs.2986#supplemental-information.

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
