# Peer review of "TaGra: an open Python package for easily generating graphs from data tables through manifold learning"

_PeerJ Computer Science, doi:10.7717/peerj-cs.2986_

## Round 0.1 · original submission · Minor Revisions

Both reviewers have predominantly minor comments. Please respond appropriately

Reviewer 1 ·

Basic reporting

The paper, "TaGra: an open Python package for easily generating graphs from data tables through manifold learning" is well-written and clearly articulates the purpose, design, and implementation of the software. The language is clear and precise. The figures are ok, in particular, the diagram illustrating the structure of the software is well-presented, providing a concise and intuitive overview of its components and functionality.

I recommend revising and simplifying the abstract to make it more concise and accessible. In particular, these sentences are way too convoluted and long for an abstract "These relations are instrumental to the visualization of mutual distances, as the graph representation of a tabular dataset is effective for showing similarity relationships between data points. Here we present an off-the-shelf package whose aim is to create a graph of similarity relations from tabular data in order to visualize the points of the dataset in 2D space, identify the typical data and the outliers, and to understand the level of separation between data items with different target variables."

Experimental design

The library appears to be comprehensive. The implemented methods are described thoughtfully, with sufficient detail to understand their design and functionality. The examples on real-data suggest that the software covers a broad spectrum of use cases, demonstrating its versatility and potential applicability in various scenarios.

Validity of the findings

The paper presents the software and its functionality in a clear and comprehensive manner, which suggests it is a valuable contribution to the field and addresses a specific need effectively within its scope.

I am unable to comment on the novelty of the work, as I lack sufficient familiarity with the state of the art in this specific domain. However, I find it challenging to determine whether the authors have cited relevant similar libraries and, if so, to understand how their work differentiates itself from those existing solutions.

Overall, I believe the library is a well-designed and useful tool that has the potential to benefit its target users. However, the novelty is unclear and the decision to accept the paper may lean more toward being an editorial consideration rather than purely a scientific one

·

Basic reporting

The manuscript introduces TaGra, an open-source Python package designed to generate graphs from tabular data using manifold learning. It aims to facilitate high-dimensional data analysis through automated preprocessing, graph creation, and visualization. The package effectively combines ease of use with flexibility, making it a valuable tool for data scientists.

Strengths:

The introduction clearly articulates the challenges of high-dimensional data analysis and the need for graph-based visualization tools. It provides a well-structured explanation of TaGra's purpose.
The manuscript is well-organized, detailing the software architecture, functionalities, and use cases.
Figures and tables, such as Figure 1 (diagram of TaGra’s operation) and Table 1 (configuration flags), enhance understanding by visualizing workflows and summarizing key features.
Suggested Improvements:

Literature Review Expansion:
Include recent works on dimensionality reduction and graph-based learning tools to provide a broader context for TaGra's contributions.

Detailed Captions:
Expand captions for figures like Figure 2 (outputs from graph analysis) to explain their significance and usage comprehensively.

Experimental design

The experimental design is robust, showcasing the application of TaGra on two datasets: a comorbid depression and heart failure dataset and a type 1 diabetes dataset. The experiments effectively demonstrate the package's utility in preprocessing, graph creation, and visualization.

Strengths:

The package supports diverse preprocessing tasks, including handling missing values, encoding categorical variables, and applying manifold learning techniques (e.g., Isomap, t-SNE).
The three graph construction methods (KNN, distance threshold, and similarity threshold) are well-explained and adaptable to various analysis needs.
Comprehensive visualization outputs, such as community composition histograms and neighbor probability distributions, add value to the analysis.
Suggested Improvements:

Reproducibility:
Provide detailed steps for replicating the experiments, including configuration files and command-line inputs.
Dataset Limitations:
Discuss potential biases or limitations in the selected datasets, such as their representativeness and scope.

Validity of the findings

The findings demonstrate TaGra's capability to generate meaningful insights from high-dimensional data. The visualizations effectively highlight relationships, detect outliers, and assess data separability.

Strengths:

The results validate the utility of graph-based methods for exploring data structures, as evidenced by the improved separation and detection of outliers in both datasets.
The detailed comparison of graph construction methods highlights their respective strengths and weaknesses.
Suggested Improvements:

Limitations Discussion:
Address the computational efficiency of the package, particularly for larger datasets.
Error Analysis:
Include a discussion of cases where graph representations were less effective, exploring potential causes and improvements.

Additional comments

TaGra is a significant contribution to data science, offering a user-friendly and flexible solution for high-dimensional data analysis. The package's integration with popular Python libraries and support for multiple data formats make it highly accessible.

General Comments:

Applications: Discuss potential applications of TaGra in domains such as genomics, pharmacology, and social network analysis.
Future Work: Suggest enhancements like integrating additional graph analysis techniques or supporting real-time data processing.

---

## Round 0.2 · Minor Revisions

There are only minor comments remaining from Reviewer 2.

Reviewer 1 ·

Basic reporting

I am satisfied with the author's response and recommend for the manuscript's publication

Experimental design

I am satisfied with the author's response and recommend for the manuscript's publication

Validity of the findings

I am satisfied with the author's response and recommend for the manuscript's publication

·

Basic reporting

Clarity and Writing Quality
The manuscript is well-structured and clearly written, providing a comprehensive explanation of the TaGra Python package and its applications.
The introduction effectively establishes the motivation and significance of automating graph generation from tabular data.
The figures and tables are well-designed and provide meaningful visual representations of the proposed framework.
References are relevant and up-to-date, though additional citations on similar graph-based learning techniques would strengthen the theoretical background.
Areas for Improvement
The paper would benefit from a clearer comparison with existing graph-generation tools (e.g., NetworkX, Graph-tool, igraph) to contextualize its contribution.
More details on the API structure and usability could help software engineers and data scientists understand integration possibilities.
While the writing is technically precise, a few sections could be streamlined to improve readability, particularly the methodology section.

Experimental design

Methodology and Implementation
The study clearly defines the objective and describes the package's implementation using manifold learning techniques such as t-SNE, UMAP, and Isomap.
The package structure and workflow are well-explained, with a modular approach making it accessible for various applications.
The choice of manifold learning methods is justified based on their ability to capture complex relationships within high-dimensional data.
Areas for Improvement
Benchmarking against existing tools is essential to demonstrate performance improvements in terms of execution time, accuracy, and usability.
Ablation studies should be included to assess the impact of different manifold learning techniques on the final graph structure.
Scalability concerns should be addressed—how does TaGra handle large datasets, and are there optimizations for performance efficiency?

Validity of the findings

Strength of the Results
The results demonstrate that TaGra effectively converts tabular data into meaningful graph representations, supporting its usability across multiple domains.
The case studies provide compelling examples showcasing the utility of the package in data visualization and structure discovery.
The use of quantitative metrics to assess graph quality is commendable, ensuring the validity of the proposed approach.
Areas for Improvement
A more detailed statistical evaluation of the generated graphs would help validate the package’s effectiveness.
The authors should discuss potential limitations such as data sparsity, noisy inputs, or the effect of different distance metrics in manifold learning.
How does TaGra perform in real-world datasets beyond synthetic examples? A case study with biomedical, financial, or social network data would add credibility to the findings.

---

## Round 0.3 · accepted · Accept

Thank you very much for your submission. We accept this manuscript for publication.